# Spatiotemporal Variation and Long-Range Correlation of Groundwater Levels in Odessa, Ukraine

**Dzhema Melkonyan** [1,2,*] and **Sherin Sugathan** [2]

1 Department of Engineering Geology and Hydrogeology, Mechnikov Odessa National University, 65082 Odessa, Ukraine
2 dScience – Centre for Computational and Data Science, Univeristy of Oslo, 0316 Oslo, Norway; sherin.sugathan@dscience.uio.no
* Correspondence: dzhema.melkonyan@dscience.uio.no; Tel.: +47-48-358-720

**Abstract:** Increasing groundwater levels (GWLs) may become one of the most serious issues for the city of Odessa, Ukraine. This study investigated the spatial distribution characteristics and multifractal scaling behaviour of the groundwater-level/-depth fluctuations for a Quaternary aquifer in the city of Odessa using a geostatistical approach and multifractal detrended fluctuation analysis (MF-DFA). These two methods were applied to monthly GWL fluctuation time series from 1970 to 2020 to monitor 72 hydrogeological wells situated in different parts of the city of Odessa. The spatial distribution of the GWLs revealed an overall trend of decline and recovery from 1970 to 2020 in the study area, except for most of the southern region, where a persistent recovery of the groundwater depth was observed. The MF-DFA results suggest that the dynamics of the GWL fluctuations have multifractal characteristics in the Odessa area. In addition, both long-range correlations and fat-tail probability distribution contribute to the multifractality. However, long-range correlations among the fluctuations made a major contribution to the observed multifractality of the GWL fluctuation time series. The generalised Hurst exponents show a wide range of change ($0.20 < h(q) < 2.85$), indicating the sensitivity of the GWL fluctuations to changes in small-scale factors and large-scale factors. Regarding the long-range correlations of the GWL depths, the Hurst exponents ($q = 2$) demonstrated the positive persistence of groundwater-depth recovery in the southern region and the persistence of groundwater-depth variation in the other regions of the study area. The dynamic changes in the GWL depths in the Odessa area may be influenced by both natural and anthropogenic factors.

**Keywords:** geostatistical methods; groundwater-level fluctuation; Hurst exponent spatiotemporal distribution; multifractal detrended fluctuation analysis

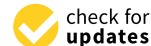



## 1. Introduction

Groundwater, located in the steppe zone of southern Ukraine, is the main water source for agriculture in the Odessa district and domestic use in the city. The formation and distribution of the groundwater in Quaternary deposits depend on the climatic, hydrogeological, geomorphological, lithological, and facies conditions, as well as human economic activity [1,2].

This study focuses on the dynamics of the shallow groundwater level (GWL), which is the depth of water in the first aquifer from the land surface. The study of GWL fluctuations is important to provide a better basis for assessing groundwater quality, quantity, and management in general, as well as to understand the dynamics and quality of the hydrogeological system and predict the processes associated with the rise and fluctuation in the GWL, such as flooding, the subsidence of loess, and landslides. At present, we are observing increasing GWLs in most residential areas of Odessa. The rising groundwater affects the basements of buildings and underground communications and leads to the subsidence of soils under foundations, which may lead to the subsidence of building foundations, potentially leading to structural damage.

In recent years, the natural regime of the Quaternary aquifers in the Odessa territory has changed radically. The water level shows a pronounced upward trend, and the area of its distribution in Odessa is also increasing. The maximum increase is confined to the central part of the city. The rise in the water level over the past 70–80 years has been 12–15 m. In most parts of the city, the water table has risen to a depth of 5 m below the land surface.

Many natural systems, the behaviours of which are outwardly perceived as chaotic, have one common property. This property is called self-affinity, also termed fractality, because nonstationary temporal fluctuations follow a power law to some extent. Numerous studies have described self-affinity in nature. These include the classic monographs of the creator of fractal geometry, Benoit Mandelbrot, and his successors [3–5], and reviews in scientific journals, books, and series [6–8]. Scientific papers devoted to the fractal properties of systems have appeared in completely different areas of natural science, including geophysics [9,10], topology [11], biology, medicine, economics, and finance. The scientific journal *Fractals* is regularly published and completely devoted to this subject. Recently, the number of publications in hydrology and hydrogeology has increased.

Various researchers have explored the scaling characteristics of groundwater systems using the detrended fluctuation analysis (DFA) method and other techniques to quantify groundwater fractal dynamics [12–14]. Li and Zhang [12] quantified the scaling properties of the nonstationary groundwater levels, streamflow, and base flow with the DFA method. Tu et al. [13] utilised DFA to quantify the monofractality, and multifractal detrended fluctuation analysis (MF-DFA) and multiscale multifractal analysis (MMA) were employed to examine the multifractal behaviour of the groundwater-level fluctuations. These results show that the multifractality of the GWL time series originated from both the long-range correlations and probability density function of the GWL series. Among other results, it was also found that crossovers of the fractals were detected for groundwater-level fluctuations [13], indicating the important role of multifractal analysis for time series. The DFA method was designed by Peng et al. [15] to investigate long-range fractal correlations in nonstationary time series. Although nonstationary fluctuations in GWLs are often characterised and modelled as fractional Brownian motions [16], the nature of GWL fluctuations is mainly nonstationary, with significant trends in GWL changes that may not be entirely random. These results indicate that, in general, groundwater-level fluctuations are formed under the influence of multiscale processes (factors) (i.e., they have a multifractal character). Many other studies have also reported similar long-term-correlated fluctuating behaviour in nature, particularly groundwater-level fluctuations [17–19]. By detrending the data using the DFA method via polynomial fittings, the impact of nonstationarity in the data can be circumvented to reveal only the fluctuations and time scaling of a process. The scaling features of GWL fluctuations established in these studies confirm that, in general, water-level fluctuations do not have a simple monofractal scaling behaviour, which can be described by a single scaling exponent, but are characterised by a more complex scaling behaviour, which can be described by several scaling exponents (multifractal behaviour). GWL fluctuations have repeatedly demonstrated long-range dependence over time [13,20–22]. These are power-law relationships over a variety of timescales that can be represented by fractals. Generally, the fractal structure of GWL fluctuations is determined by a power-law exponent based on the assumption that the scaling is independent of space and time in the DFA. Thus, literature reviews suggest that GWL fluctuations are influenced by multiscale processes (factors) (i.e., they have a multifractal character).

The existence of crossover timescales with different scale exponents cannot be reliably and authentically characterised by a determined mathematical form. In this regard, DFA cannot describe the detailed behaviour of fractal scales. Multifractal DFA (MF-DFA) was introduced to overcome the limitations of DFA [23]. As an effective tool for fractal analysis, the MF-DFA method has been successfully applied to analyse complex phenomena in various areas of science, as well as to quantify the fractal dynamics of groundwater systems [12,13,18,22]. The local Hurst exponent [24] is a useful tool for characterising the

multifractal behaviour of GWL fluctuations affected by variations in different factors in space and time. Moreover, it sheds light on the prediction of GWL fluctuation trends.

Understanding the spatiotemporal changes in the groundwater levels is of great importance for hydrogeological forecasting in the Odessa territory, and for understanding the mechanisms of the processes and phenomena associated with GWL fluctuations. In addition, variations in the water level, as well as other parameters characterising the Quaternary aquifer, reflect the properties of groundwater to receive and transmit signals regarding changes in the stress–strain characteristics; the filtration characteristics of the aquifer rock are influenced by different factors.

The MF-DFA method proves that groundwater-level fluctuations are formed under the influence of multiscale processes (factors) and makes it possible to predict GWL fluctuation trends. Geostatistical methods allow for separating these multiscale factors (structural, natural, and local) and determining their role in GWL fluctuation. Therefore, in this study, we combined the MF-DFA method and the geostatistical approach to analyse the spatial distribution characteristics of the GWL depths and the multifractal features of a long-term monthly GWL time series of the Quaternary aquifer recorded in the Odessa area from 1970 to 2020. The main objectives of this study were to predict the GWL depths in the city of Odessa using a long-range fractal correlation approach, and to reveal how the large-scale and small-scale factors influence the spatial correlation of the GWL depths.

## 2. Materials and Methods

### 2.1. Site Description and Data Collection

The study area is located on the north-western shore of the Black Sea (Odessa, Ukraine; Figure 1), where Quaternary deposits are widespread. The groundwater in the Odessa territory is mainly concentrated in the Quaternary strata. The climate of the Odessa region is continental moderate. The mean annual temperature in the region is approximately 10.6 °C, with a summer mean of +22.6 °C and a winter mean of −0.5 °C. The mean annual precipitation ranges from 432 to 453 mm. The Odessa region lies within the Black Sea lowland. The region is composed of Meotian, Pontian, Middle–Upper Pliocene, and Pleistocene formations [2]. These formations gently dip uniformly in a southerly direction and are approximately perpendicular to the coastal slope. Meotian strata (clays) are exposed along the Odessa Bight. The Meotian clays are overlain by Pontic limestones and sandy and clay deposits of Upper Pliocene and Pleistocene loesses [2].

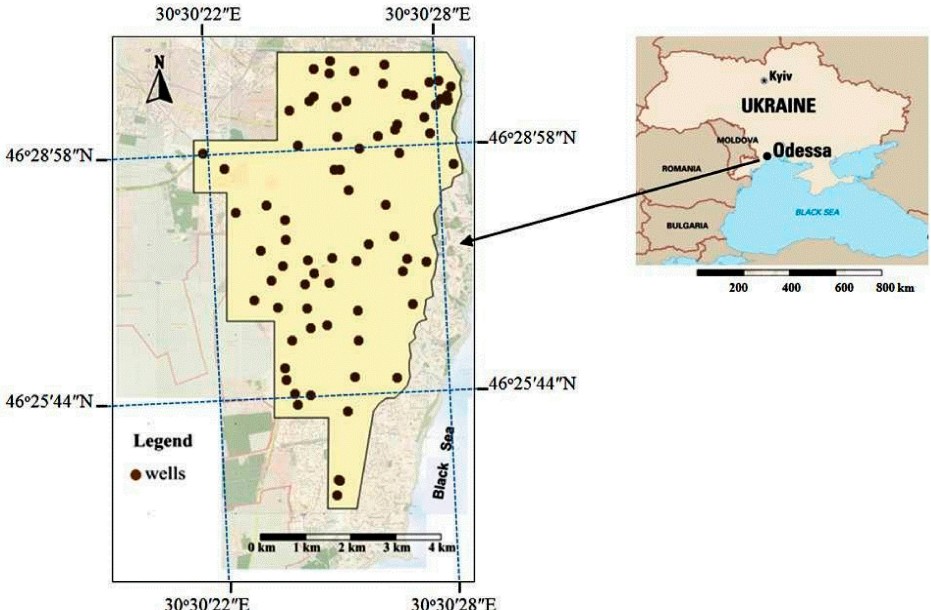

**Figure 1.** Black Sea and location of Odessa. Location of groundwater-monitoring wells in territory of Odessa.

The Odessa region is located within the Black Sea artesian basin. The aquifers that are exploited in the region are in Quaternary loess sediments, Pontian limestones, and Sarmatian sand, limestone, and grit-stone sediments. The Sarmatian aquifer lies below the Meotian clays. The upper horizons and complexes are recharged mainly via precipitation and are exposed by erosion incisions. Over the last century, due to dredging activity by seaports, water leakages from the infrastructure in urban areas, and excessive irrigation in agricultural areas on the coast, the share of irrigation water and leakage from water supply lines has increased in Quaternary deposits. Within these strata, two or three aquifers are typically observed. The aquifer sediments are composed mainly of loess and loessoid loams. The thickness of the water-bearing layer in the Odessa city area varies from 0.2 to 5.9 m. The groundwater level is recorded at depths of from 3.0–5.0 m to 8.0–12.0 m, and at depths of 0.5–2.5 m in flooded areas. The aquifer recharge is driven mainly by atmospheric precipitation and artificial recharge. The groundwater discharges onto the slopes of ravines, gullies, estuaries, and seashores. The median depths of the Quaternary water wells are in the range of from 12 to 40 m. We used an unconfined water mirror. The groundwater levels were taken directly from measurements.

Groundwater can accelerate the loess slope stability in the city of Odessa [25,26]. The intensity of sliding displacements on the loess slopes in the Odessa territory increases with the appropriation of new territories, mainly because of the rising GWL in the Quaternary aquifer [1,2]. Loess rocks have poor filtration properties, with filtration coefficients typically between 0.1 and 1.0 m/day. Loessoid loams have pronounced filtration anisotropy, which is expressed by significantly greater value of filtration coefficients in the vertical direction (2–8 times more) than in the horizontal direction [2,26]. Therefore, precipitation and wastewater move to the aquiclude quicker than they spread horizontally. The downward movement of the groundwater into underlying aquifers mainly occurs in places where Pliocene reddish-brown clays underlying the loess layer are absent, in areas adjacent to gullies and ravines, and in areas of coastal cliffs via landslide bodies [27]. Part of the Quaternary water is discharged into the Pontian sediments through the drainage system.

This research was conducted using data provided by the Department of Engineering Protection of the Territory of Odessa and the Department of Engineering Geology and Hydrogeology of Mechnikov Odessa National University.

Monthly GWLs were measured over a 50-year period (starting in 1970) for 72 monitoring hydrogeological wells located on an area of 25.8 km$^2$ in the city of Odessa. The Newton interpolation method was used to interpolate the GWL time series for short-period missing data. The GWL occurred at a depth of 3–6 m (4.5 m on average) for the largest area. The total area of these zones is 17.33 million m$^2$, or 68.07% of the total area occupied by hydrogeological well monitoring. The area of zones with GWL depths of less than 3 m (average of 2.5 m) was 13.28 million m$^2$ (13.28%). In 18.65% of the study area, the GWLs did not reach depths of less than 6 m.

## 2.2. Method Description

In this study, the spatial variability and correlation of the GWL depths in the city of Odessa were studied using geostatistical methods in Golden Software Surfer 13. The MF-DFA method was used to examine the multifractal behaviour of the GWL fluctuations, and to analyse the temporal variability of the GWL depths and long-range fractal correlations.

### 2.2.1. Geostatistical Methods

To consider the spatial correlation between the measured data, we used geostatistical methods, which included semivariogram functions and kriging interpolation. Many researchers have used these methods to identify the spatial and temporal structures of GWL depth fluctuations [28–30]. In this study, basic geostatistical methods were used to quantify the spatial variability of the GWL depths in the study area. Ordinary kriging (OK) was used as the interpolation method. For ordinary kriging, the data series should have a normal or lognormal distribution [31]. We tested the normality of the data using

the Kolmogorov–Smirnov (K-S) test and Shapiro–Wilk (S-W) tests. As the GWL datasets followed a lognormal distribution, a log transformation was carried out to normalise the time-series data. Descriptive statistics were examined, including the mean, median, coefficient of variation, skewness, and kurtosis. A semivariogram was used to quantify the differences between the sampled data values as a function of the separation distance.

A semivariogram plot was obtained by calculating the semivariograms at different lag distances. The generalised formula of the semivariogram function is as follows [32,33]:

$$\gamma(h) = \frac{1}{2N(h)} \sum_{i=1}^{N(h)} [(z(x_i) - z(x_i + h)]^2 \tag{1}$$

where $\gamma(h)$ is the semivariogram value at the distance $h$; $z(x_i)$ is the sample value at the spatial point $x_i$; $N(h)$ is the total number of the variable pairs separated by the distance $h$. For a quantitative evaluation of the variation characteristics of the research area, determining the theoretical model of its semivariogram is necessary. Common semivariograms are generated using different models (exponential, spherical, and Gaussian) [34].

The semivariogram function of the GWL in the city of Odessa can be fitted with a Gaussian function (Equation (2), parabolic shape):

$$\gamma(h) = \begin{cases} 0 & h = 0 \\ C_0 + C\left(1 - e^{-\frac{3h^2}{a^2}}\right) & h > 0 \end{cases} \tag{2}$$

The three main parameters of the semivariogram are $C_0$, $C_0 + C$, and $a$. Here, $C_0$ is the localised discontinuity, or nugget, which is the semivariogram function value as a result of the measurement error and spatial variation at distances shorter than the smallest sampling interval [30]; $C_0 + C$ is called the sill, the relatively stable value that the semivariogram reaches if the distance between the sampling points increases by $h$; $a$ is the distance between the sampling points when the semivariogram reaches the sill from the nugget. This distance describes the range of the spatial correlation, and $C$ is the difference between the sill and nugget.

A fitting diagram of the semivariogram function of the groundwater depth in the city of Odessa for the turning years is shown in Figure 2. The nugget and sill can be used to characterise the degree of spatial variability in groundwater depths [30]. The nugget denotes the variation as a result of random small-scale factors, including human factors, such as local leaks from water-bearing utilities, local heavy precipitation, and local irrigation. The sill represents large-scale variations as a result of natural factors, including the topography, geology, climate, and sea level, and large-scale human factors, such as the large-scale exploitation of groundwater. In general, the nugget/sill ratio ($C_0/(C_0 + C)$) can be used to classify the spatial dependence. The ratio of these two parameters (also called the nugget effect) can exhibit the strength of the spatial correlation of GWL depths [30,35].

We denote this ratio as $C_{ne}$ and present it in the following form:

$$C_{ne} = \left(1 + \frac{C}{C_0}\right)^{-1}. \tag{3}$$

If $C_{ne} < 25\%$, the variables are highly affected by natural factors and strongly correlated in space; if $25\% \leq C_{ne} \leq 75\%$, the variables are influenced by both natural and random factors and moderately correlated in space; and when $C_{ne} > 75\%$, the variables are extremely influenced by random factors and have a low spatial correlation. When $C_{ne}$ is higher, and the spatial correlation is smaller, it is obvious that the spatial variability is the result of stochastic factors. A $C_{ne}$ close to one suggests the constant variability in the variable on a small scale [36].

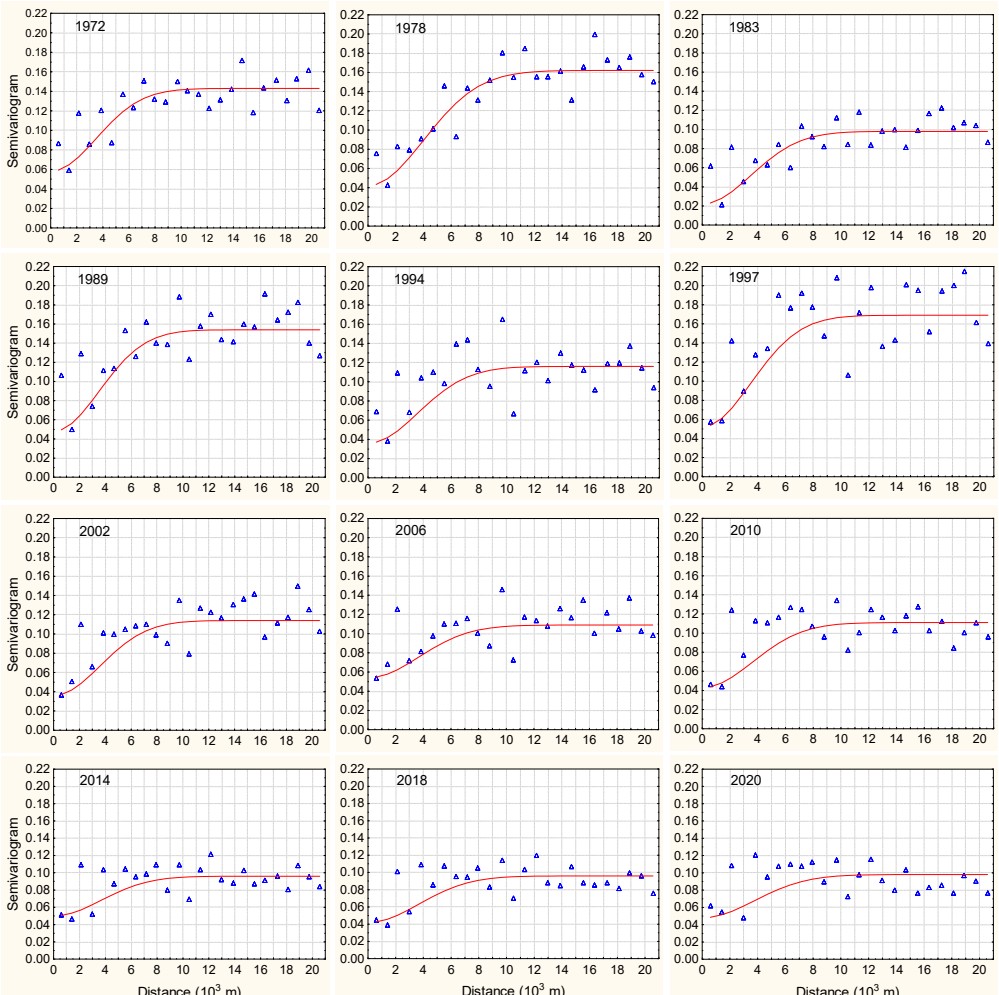

**Figure 2.** Semivariogram functions of groundwater-level depths in the city of Odessa (Gaussian model).

The universal kriging method with cross-validation was applied to find out whether the dataset followed any trend, and to assess the accuracy of the chosen variograms in the estimation of the GWL fluctuations for the spatial scale.

As noted by Lu et al. [30], rather than assuming that the mean groundwater depth is constant over the entire domain, we recognise that it may contain a spatial trend, which is appropriately interpolated via the OK procedure. The parameters for the variogram models were derived following the step-by-step procedures outlined by Barnes [37], and an initial theoretical variogram was fitted to the experimental variogram using the Surfer computer program's interactive interface. The high $R^2$ values obtained from the kriging estimations proved that the variograms were chosen correctly.

### 2.2.2. Multifractal Detrended Fluctuation Analysis (MF-DFA)

There are two types of processes with fractal properties: monofractal (self-similar or self-affine) and multifractal. Multifractals are often found in nature, whereas simple self-similar objects, such as monofractals, represent an idealisation of real phenomena. Monofractal processes are homogeneous because their scaling characteristics remain unchanged over any scale range. Multifractal processes are characterised by different scaling behaviours (or by a spectrum of scaling exponents) of different moments over a full range of timescales.

MF-DFA is a generalisation of DFA, developed by Peng et al. [15], and it is used to detect long-term correlations in nonstationary time series. Based on DFA, MF-DFA was

developed by Kantelhardt et al. [23] for the multifractal characterisation of nonstationary time series. The MF-DFA method has been applied to evaluate the characteristics of data, such as long-term weather records [38], geology time series [39], neuron spiking, and financial time series. Although several studies on the fractal features of GWL fluctuations have been published in recent years [12,13,18,22], the MF-DFA method has not been sufficiently tested for the analysis of GWL fluctuations. Moreover, the application of this method to the study of the temporal and spatial variations in the GWL in Odessa is relatively rare.

The most important step in the execution of both DFA and MF-DFA is eliminating the local polynomial trend ($Y(j)$) in each segment of the cumulative series ($X(t)$) of length $N$.

Basic MF-DFA consists of the following steps [10,23,40–42]:

1. The stochastic time series $x(t)$, $t = 1, 2..., N$ is shifted by the mean $\langle x \rangle$ and cumulatively summed:

$$X(t) \;=\; \sum_{i=1}^{t} [x_i - \langle x \rangle].$$

(4)

2. The integrated time series $X(t)$ is segmented into $N$ non-overlapping windows of equal size ($s$ ($N_s = N/s$)). Because the length ($N$) of the time series is not always a multiple of the considered time scale ($s$), a short part at the end of the series may remain. To avoid disregarding this part of the data, the same procedure is repeated starting from the opposite end. As a result, we obtain $2N_s$ segments altogether;

3. For the calculation of the trend of each $2N$ segment, a least-square fit method is applied, and the variance is determined as follows:

$$F^2(v,\,s) \;=\; \frac{1}{s} \sum_{j=1}^{s} \{Y[(v-1)\,s+j] - Y_v(j)\}^2.$$

(5)

for each segment ($v$), $v = 1, \ldots, N_s$, and as follows:

$$F^2(v,\,s) \;=\; \frac{1}{s} \sum_{j=1}^{s} \{Y[N - (v - N_s)\,s + j] - Y_v(j)\}^2$$

(6)

for $v = N_s + 1, \ldots, 2N_s$. In Equations (5) and (6), $Y_v(j)$ is a locally fitted polynomial of the first order or higher. In this study, local trends were fitted using a first-order polynomial;

4. The fluctuation function of the $q$th-order and $2N_s$ segments is determined as follows:

$$F_q(s) \;=\; \left\{ \frac{1}{2N_s} \sum_{v=1}^{2N_s} [F^2(v,\,s)]^{q/2} \right\}^{1/q}$$

(7)

where $q \neq 0$ and $s \geq m + 2$ ($m$ is the order of $Y_v(j)$ to be fitted). In this study, $m = 2$, and $s = N/4$. To reveal how the function $F_q(s)$ depends on the $s$ for different $q$ values, we need to repeat Equations (5)−(7) for $s$ different timescales;

5. Finally, the scaling behaviour of the fluctuation functions was determined by analysing log–log plots of $F_q(s)$ versus $s$ for each value of $q$. If a long-range power-law correlation exists in the time series, the $F_q(s)$ for large values of $s$ increases according to the power law [23]:

$$F_q(s) \;\sim\; s^{h(q)}.$$

(8)

In Equation (8), $h(q)$ is a generalised Hurst exponent and takes a variety of values for each value of $q$. For stationary time series, the exponent $h(2)$ is identical to the classical Hurst exponent $H$ and varies between 0 and 1 [3], and $H = h(2) - 1$ for nonstationary time series. For $q > 0$, $h(q)$ depicts the scaling behaviour of the segments with large fluctuations, and for $q < 0$, $h(q)$ describes the scaling behaviour of the segments with small fluctuations [24].



Because Equation (7) contains uncertainty at $q = 0$, the following expression is preferred for $F_q(s)$:

$$F_0(s) \; = \; \exp\left\{\frac{1}{2N_s}\sum_{v=1}^{2N_s}\ln[F^2(v,\,s)]\right\} \; \sim \; s^{h(0)} \tag{9}$$

The generalised Hurst exponent ($h(q)$) is independent of the $q$ for a time series of monofractal behaviour and strongly depends on the $q$ for a time series showing multifractal behaviour, which means that the scaling behaviour differs for fluctuations of different magnitudes. The Hurst exponent ($h(q)$) is interpreted as follows [40]: $h \in (0, 0.5)$ indicates the antipersistency (or short-term persistence) of the time series. In this case, growth in the past means a decrease in the future, and a downward trend in the past makes an increase in the future likely; $h = 0.5$ indicates uncorrelated noise, indicating no correlation between the past and future for all $t$, as it should be for a random process with independent increments indicating the absence of long-term correlations; $h \in (0.5, 1)$ indicates the persistency of the time series, meaning that the time series is long-range positively correlated (that is, an increase is more likely to be followed by an increase, and vice versa); $h = 1.5$ indicates Brownian motion (integrated white noise); $h \geq 2$ indicates black noise.

The Hurst exponent ($h(q)$) relates to the classical multifractal scaling exponent ($\tau(q)$) through the following relation:

$$\tau(q) = q\, h(q) - 1. \tag{10}$$

The transition from the Hurst exponent ($h(q)$) to the main characteristics of multifractals, such as the singularity index ($\alpha$) that represents the local Hurst exponents and the spectral function ($f(\alpha)$) of the time series, can be carried out via Legendre transform:

$$\alpha(q) = d\,[\tau(q)]/dq, \tag{11}$$

$$f(\alpha) = \alpha\, q(\alpha) - \tau(q(\alpha)). \tag{12}$$

The singularity index ($\alpha$) quantifies the singularity intensity, and the $f(\alpha)$ represents a multifractal spectrum of the time series. This spectrum illustrates the deviation of segments with small and large fluctuations from the average fractal structure. The width of the spectrum is a direct measure of the degrees of multifractality, which reflects the strength of the multifractality effects in the time series. This strength can be estimated by the width of ($\alpha$), which is defined as the difference between the maximum and minimum values of ($\alpha$): $\Delta\alpha = \alpha_{max} - \alpha_{min}$. The wider the spectrum, the richer the multifractality behaviour of the analysed time series. Usually, $f(a)$ is a smooth upper convex curve. Each point on the $f(a) \sim a(q)$ curve represents the fractal dimension of the subset with the same singular exponent ($a(q)$).

For the MF-DFA, this study used a Python-based implementation of MF-DFA introduced by Gorjão et al. [43]. They have provided an open-source implementation at https://github.com/LRydin/MFDFA (accessed on 30 September 2022).

## 3. Results and Discussion

### 3.1. Groundwater-Level Spatiotemporal Characteristics

As a preliminary investigation, graphs of the average monthly variations in the GWLs are shown in Figure 3 for selected wells: 198, 199, 1349, 247, 1634, and 1637. All 72 monitoring wells had at least 3198 water-level records. The wells exhibited nonstationary GWL changes. As shown in Figure 3, during the observation period of 1970–2020, the greatest increases in the groundwater in the territory of Odessa took place at the following times: 1977–1978; 1988–1989; 1997–1998; 2005–2006; 2010–2011; and 2015–2016.

Figure 4 shows the average GWL depths with the corresponding seasonal distributions. The monthly boxplots show that, in the average annual cycle, the water level peaks mainly in April and May and decreases in September and October. The rates of change vary from 0.9 to 0.2 m/year (mean = 0.19 m/year).

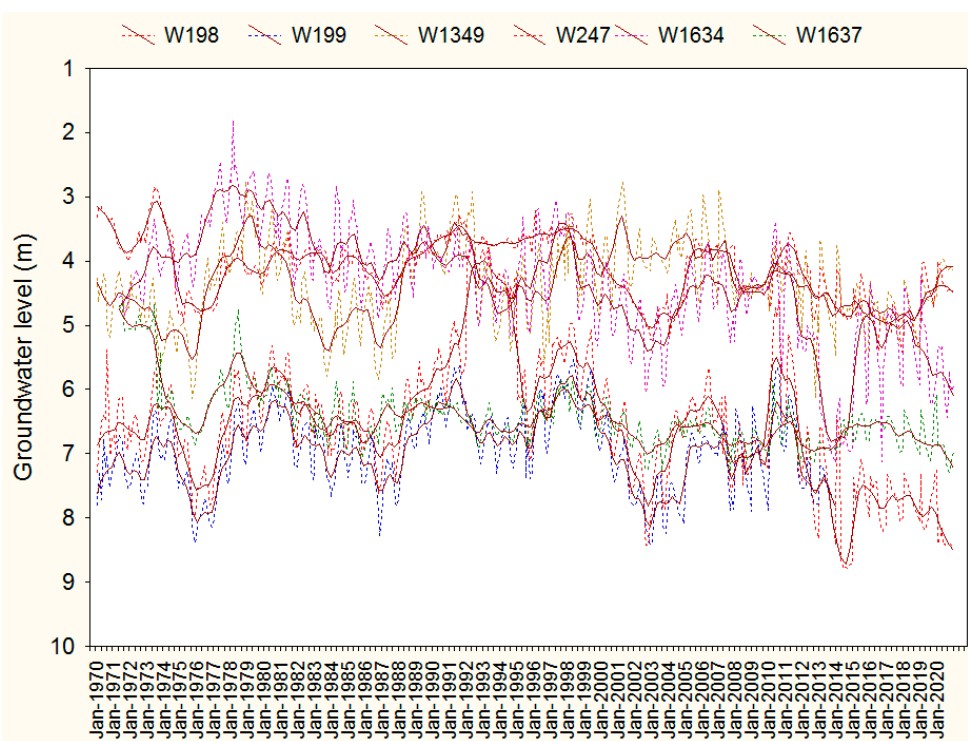

**Figure 3.** Inter-annual dynamics of groundwater levels in monitoring wells (198, 199, 1349, 247, 1634, and 1637) in the territory of Odessa.

The K–S single-sample test showed that the logarithm of the GWL data was normally distributed. Table 1 summarises the statistics of the data from 1970 to 2020 for the turning years of GWL fluctuations: 1972, 1978, 1983, 1989, 1994, 1997, 2002, 2006, 2010, 2014, 2018, and 2020. The mean and median groundwater depths increased and then decreased with time. This pattern characterises the general trend of recovery and then decline in the GWL depths in the city of Odessa from 1970 to 2020.

**Table 1.** Statistical parameters of the groundwater-level depth (in meters) in the Odessa city area.

| Year | Mean | Median | Variation | Kurtosis | Skewness |
|------|------|--------|-----------|----------|----------|
| 1972 | 5.09 | 4.54 | 2.50 | 0.49 | 1.06 |
| 1978 | 4.25 | 3.77 | 1.94 | 0.46 | 2.10 |
| 1983 | 4.75 | 4.36 | 1.91 | 0.40 | 2.19 |
| 1989 | 4.49 | 3.91 | 2.04 | 0.46 | 3.03 |
| 1994 | 4.68 | 4.44 | 1.86 | 0.40 | 3.01 |
| 1997 | 4.04 | 3.73 | 1.89 | 0.47 | 2.35 |
| 2002 | 4.93 | 4.87 | 1.81 | 0.37 | 2.63 |
| 2006 | 4.51 | 4.15 | 1.77 | 0.39 | 2.94 |
| 2010 | 5.03 | 4.54 | 2.03 | 0.40 | 2.59 |
| 2014 | 5.80 | 5.49 | 2.02 | 0.35 | 3.91 |
| 2018 | 5.22 | 5.10 | 1.61 | 0.31 | 0.46 |
| 2020 | 5.77 | 5.74 | 1.63 | 0.28 | 0.78 |

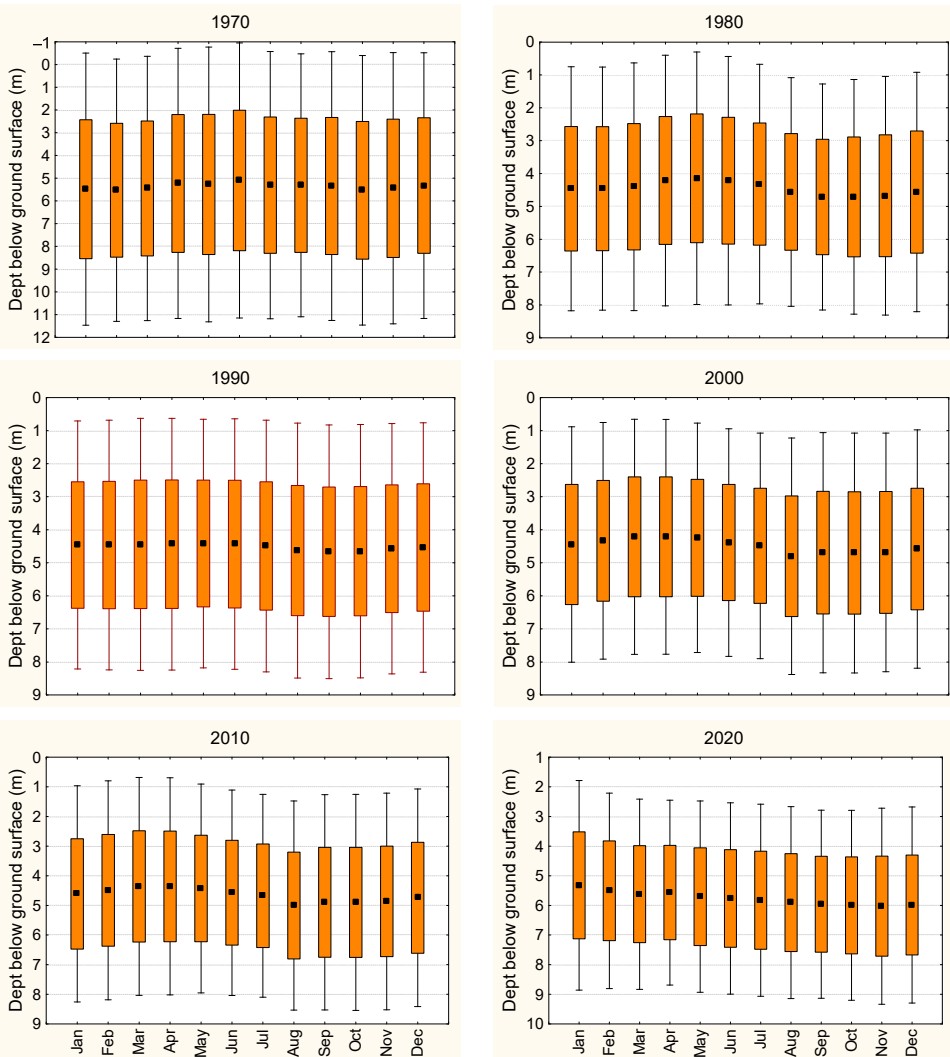

**Figure 4.** Boxplots of monthly groundwater-level distributions for 72 monitoring wells (from January to December of 1970, 1980, 1990, 2000, 2010, and 2020).

The characteristic depths of the groundwater in the studied territory were not the same in different parts of the city. Therefore, maps to show the spatial distributions of the GWL depths in different years were necessary. We selected the turning years of GWL fluctuations. The maps for these years are presented in Figure 5, after ordinary kriging interpolation [44], and they show the spatiotemporal evolution of the average GWL depth in the study area.

As shown in Figure 5, the study area was divided into three regions: northern, central, and southern. The boundaries between the regions are indicated by red dotted lines. Overall, during the past decade, the GWL has been shallower in the central and southern regions. In these regions, the average groundwater depth was 5–6 m or less. In some cases, it reached up to 3 m, such as in the western part of the central region and in the central part of the southern region. In the northern region, the GWL also became shallower than in the central and southern regions, although it is not as pronounced.

In this study, we used the semivariogram model parameters in order to examine the spatiotemporal variability characteristics of the GWL depth correlations. These parameters are shown in Figure 6, including the nugget ($C_0$), sill ($C_0 + C$), and ratio of the nugget to the sill ($C_{ne}$). It is noteworthy that the $C_0$ and $C_0 + C$ reflect factors affecting the groundwater depth. The parameter $C_0$ reflects the impact of small-scale factors, which include small-scale human factors, particularly local irrigation practices and leakage from water-bearing utilities,

and climatic factors, particularly local heavy rainfall. The parameter $C_0 + C$ reflects the impact of large-scale structural factors, which include the sea level, topography, climate, and geology (aquifer lithology), as well as large-scale human factors, particularly activity by seaports, the large-scale construction of water conservancy establishments, and large-scale mining.

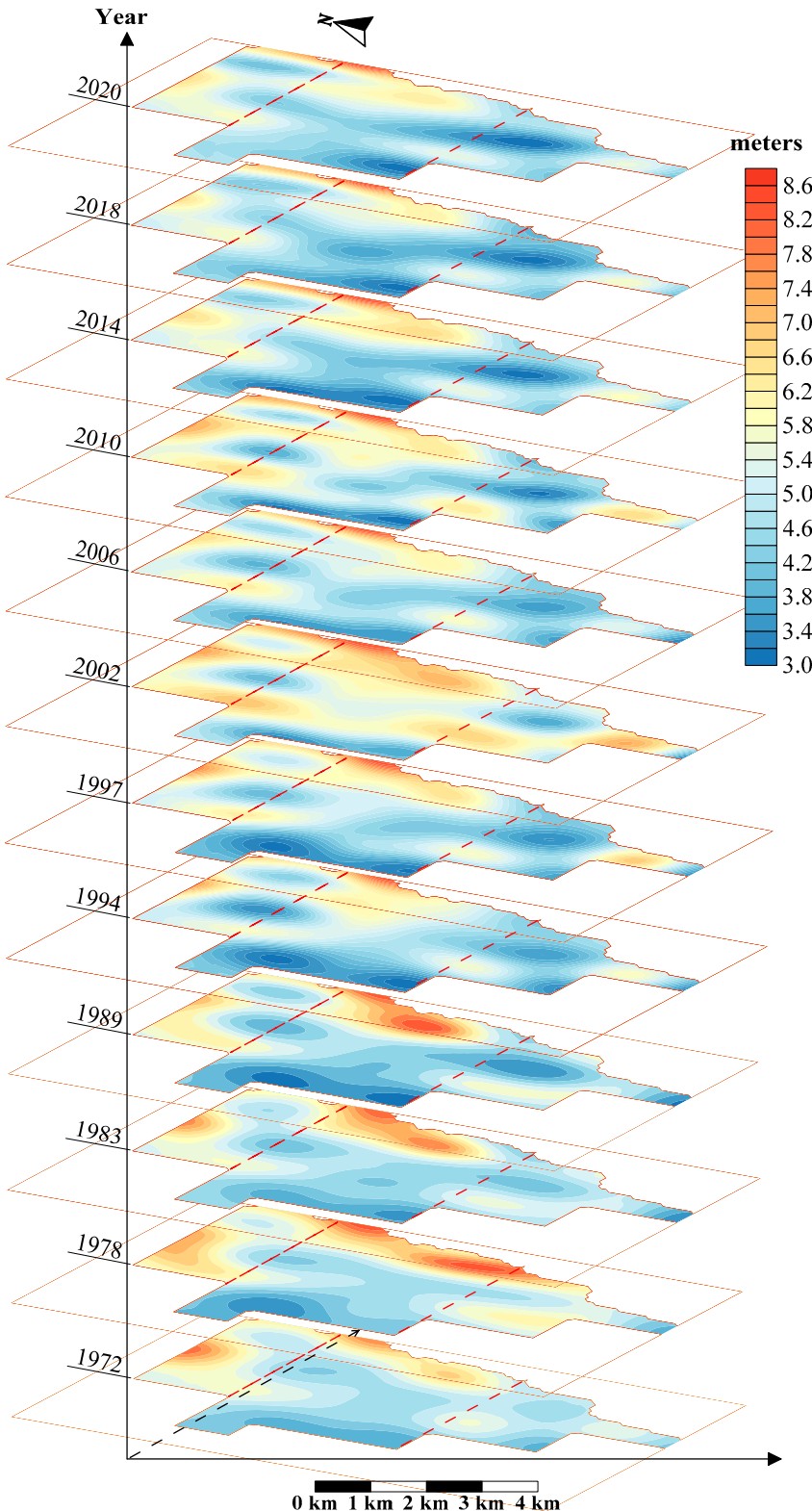

**Figure 5.** Spatiotemporal distribution of groundwater-level depth in the Odessa city area. The red dotted lines are boundaries between the northern, central, and southern regions.

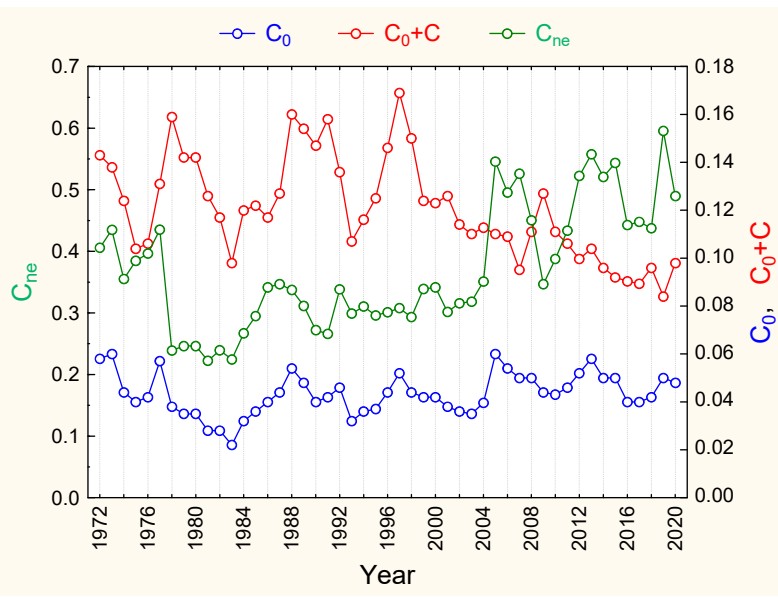

**Figure 6.** Semivariogram-function analytical model and related characteristics of groundwater-level depths in Odessa.

The turning years of 1972, 1978, 1983, 1989, 1994, 1997, 2002, 2006, 2010, 2014, 2018, and 2020 were selected to analyse the dynamic changes in the GWL depth in the Odessa city area. As shown in Figure 6, the ratio of the nugget to sill ($C_{ne}$) was greater than 25% (between 35% and 43%) and approached 25% over the period 1972–1983, which indicates that the spatial correlation of the GWL depths in the study area moved from a moderate to a strong correlation. Meanwhile, the $C_0 + C$ value remained moderate, whereas the $C_0$ value remained low, which indicates that the GWL showed an overall change in different parts of the study area. As seen in Figure 5, the spatial correlation of the GWL changes increased in 1972, 1978, and 1983. According to these graphs, the cone of the depression continued to expand to the eastern area of the central region, and the groundwater depth continued to increase. In contrast, the groundwater rebounded in the western area of the central region and in the central and southern areas of the southern region. This overall variation indicated that the change in the GWL depths over this period was mainly caused by a combination of natural and human factors. It is obvious from Equation (3) that the $C_{ne}$ can decrease due to both a decrease in the $C_0$ and an increase in the $C$. Based on this, and from the graphs shown in Figure 6, one can see that, during the period 1972–1983, strong spatial correlations appeared due to a decrease in the $C_0$ (a trend can be observed) and a partial increase in the $C$ (a trend is not observed). In addition, decreases in the local factors can lead to an increase in the role of structural factors, such as the sea level, climate, and geology. Figure 6 shows that the variation character of the $C_{ne}$ is more consistent with the variation character of the $C_0$.

During 1984–1990, the $C_{ne}$ value increased (35%) and then decreased (26%), indicating a strong correlation between the GWL depths in different parts of the study area during this period. This indicates that the GWL depths showed an overall change during this period. As shown in Figure 5 (1989), the GWL depths in the eastern part of the central region increased, and they rebounded in the western part of this region and in the central part of the northern and southern regions. In addition, Figure 6 shows that, during this period, both the $C_0$ and $C_0 + C$ values increased; however, the $C_0 + C$ increased faster than the $C_0$. In this case, according to Equation (3), a strong correlation is ensured by increasing the role of the structural factors.

Notably, during 1972–2004, the three parameters $C_0$, $C_0 + C$, and $C_{ne}$ exhibited relatively synchronous fluctuations.

During 1991–2004, the $C_{ne}$ value gradually increased and approached 35%. As shown in Figure 5 (graphs for 1994, 1997, and 2002), the groundwater cone of the depression continued to expand and the GWL depths continued to increase in all regions. In addition, the $C_0$ remained low and almost unchanged. The $C_0 + C$ value remained high and decreased during this period, which led to an increase in the $C_{ne}$ and a weakening of the spatial correlation. However, the $C_0 + C$ value remained high, indicating that the GWL depth variation was affected by natural factors (the sea level, topography, climate, aquifer lithology, and different hydrogeological units) and human factors (continuous leaks from water-bearing utilities, activity by seaports, various waste filtration reservoirs, and large-scale mining).

During 2005–2020, the $C_{ne}$ fluctuated between 35% and 60%, specifying a moderate spatial correlation with the GWL depths in different areas of Odessa. The $C_0$ remained almost unchanged (Figure 6) and the $C_0 + C$ decreased, which, according to Equation (3), led to a decrease in the $C_{ne}$ and weakened the spatial correlation. The five GWL depth graphs for 2006, 2010, 2014, 2018, and 2020, plotted in Figure 5, show slight decreases in the GWL depths in all regions of the study area. The $C_0$ and $C_0 + C$ also reflect that, since 2005, the GWL depths in Odessa may have been affected by natural factors (the climate, sea level, topography, and aquifer lithology) and human factors (activity by seaports, leaks from water-bearing utilities, and various waste filtration reservoirs).

### 3.2. Groundwater-Level Multifractal Characteristics

In this study, we analysed the monthly water-level records for 72 monitoring wells using MF-DFA analysis for multifractal behaviour. The multifractal results include log–log plots of the $F_q(s)$ against the timescale ($s$), the generalised Hurst exponent ($h(q)$), the scaling exponent spectrum ($\tau(q)$), and the singularity spectrum ($f(\alpha)$) corresponding to a series of moments ($q$ ($-10 \leq q \leq 10$)). In Figure 7, we present the plots $h(q)$ vs. $q$, $\tau(q)$ vs. $q$, and $f(\alpha)$ vs. $\alpha$ for several wells. Our analysis shows that multifractal behaviour exists for the GWL fluctuations in all the wells, as the $h(q)$ varies with the $q$ and the relationships between the $\tau(q)$ and $q$ are not linear, as demonstrated in Figure 7a,b for several wells.

The generalised Hurst exponent ($h(q)$) is a fluctuation parameter that describes the correlation structures of the time series at different magnitudes. Figure 7a shows that the $h(q)$ continuously decreases as the $q$ increases, reflecting the fact that relatively small fluctuations occur more frequently in the time series than large ones [45]. The analysis of the GWLs shows that, for all the considered wells, the generalised Hurst exponent ($h(q)$) has a wide range of change of $0.20 < h(q) < 2.85$. Such a wide range of change in the $h(q)$ indicates an increase in the multifractal properties and the appearance of long-range correlations in the GWL time series. From a hydrogeological perspective, this is expressed by the strong sensitivity of GWL fluctuations to changes in small-scale factors, including human factors (local leaks from water-bearing utilities, local heavy precipitation, and local irrigation) and large-scale factors (the topography, geology, climate, and sea level, and large-scale human factors, such as activity by seaports, large-scale mining, and irrigation).

The singularities of the processes at the GWL depths of the considered wells are shown in Figure 7c. The width of the singularity spectrum ($\Delta\alpha$ ($\Delta\alpha = \alpha_{max} - \alpha_{min}$)) measures the degree of multifractality. The wider the singularity spectrum, the stronger the multifractal features. For all the considered wells, the $\Delta\alpha$ was found to be within 0.23–2.60, except for three wells, for which it was found to be 4.16, 4.26, and 4.47. This indicates a high degree of multifractality and suggests that the groundwater multifractal behaviour is quite specific.

Multifractal structures can be caused by the long-term correlations of different fluctuation processes and fat-tail probability distributions of the time series [23,45]. The random shuffling method was used to analyse the sources of multifractality. If the multifractality is due to both types, the shuffled data present weaker multifractality than the original data. For less than half of the wells, the shuffled data showed weaker multifractality, indicating that both types of multifractality were present in the time series.

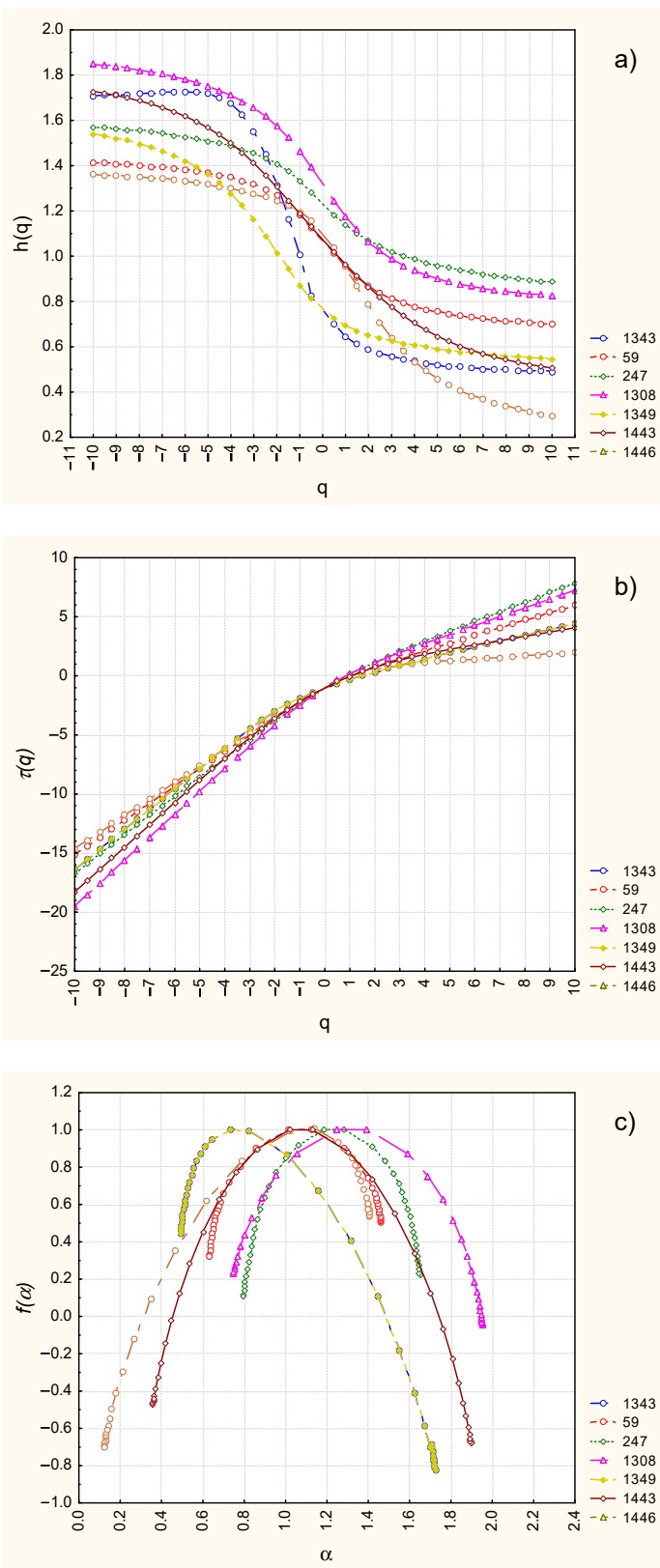

**Figure 7.** Multifractal analysis for groundwater time series of wells 1343, 59, 247, 1308, 1349, 1443, 1446: (**a**) generalised Hurst exponent (*h*) as a function of *q* for the groundwater-level depths; (**b**) multifractal scaling exponent spectrum ($\tau(q)$); (**c**) singularity spectrum ($f(\alpha)$).

### 3.3. Hurst Exponent Spatial and Temporal Distribution and Groundwater-Level Prediction

The Hurst exponents ($h = 2$) of the GWL fluctuation data quantified using the MF-DFA approach for the 72 monitoring wells were investigated. The Hurst exponent at $q = 2$ is identical to the classic Hurst exponent of a monofractal stochastic process. This study used a sliding time series to quantify the Hurst exponent by considering 480 monthly GWL depth data points as the length of the time series.

Sliding was performed in a sequence of 40 years with a step length of 1 year. The first Hurst exponent was defined using GWL time series for the period $1970-2009$, the second Hurst exponent was estimated for the period $1971-2010$, and so on. The contour maps of the Hurst exponent for the studied GWL time series are shown in Figure 8.

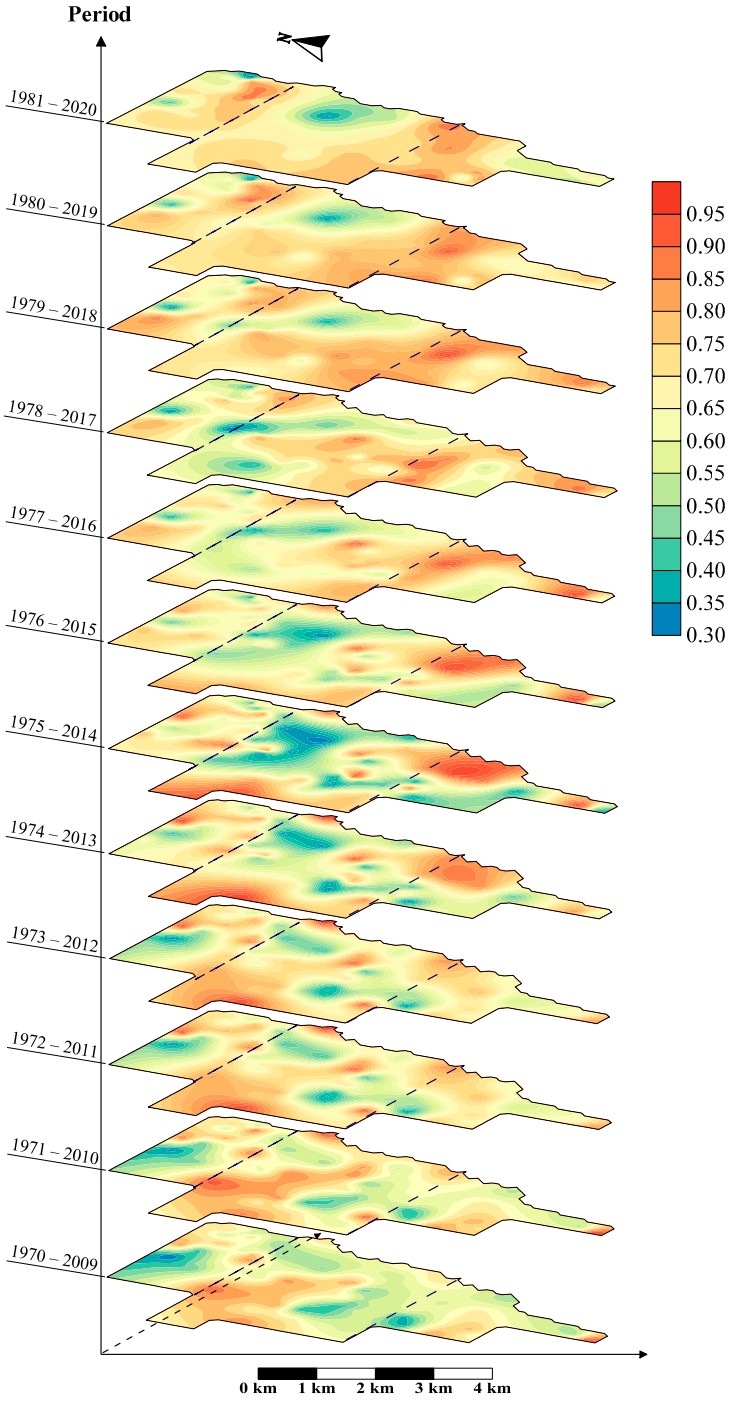

**Figure 8.** Contour maps of the Hurst exponent ($h = 2$) in the territory of Odessa.

The higher the Hurst exponent value, the stronger the memory, the longer it will remember previous values, and the more consistent its ability to maintain the previous change trend [30,46]. Moreover, an increase in the sliding Hurst exponent value specifies that the time series can remember the later GWL depth trends better. A decrease in the Hurst exponent indicates that the time series remembers earlier GWL depth trends better.

Figure 8 shows that the Hurst exponent value changed within the regions of the study area, showing varying degrees of long-range correlation. Thus, in most parts of the study area, the Hurst exponent exceeded 0.5, indicating that the GWL fluctuations exhibit persistent behaviour and may show a stable trend, and the long-range correlation of the GWL depth is predicted to be significant. In addition, an increase in the Hurst exponent indicated that the overall GWL depth in most parts of the study area decreased. The accuracy of the Hurst exponent was confirmed via the spatiotemporal distribution of the GWL depth in the study area (Figure 5).

Thus, by comparing the maps of the spatial distribution of the GWL depth (Figure 5) and the Hurst exponent (Figure 8), we can roughly determine that, for the eastern and north-western parts of the northern region, a slight recovery of the groundwater may occur in the coming period. For other parts of the northern region, the groundwater is expected to continue to increase in the coming years. In most parts of the central region, the groundwater may maintain a continuous rebound trend. In the eastern part of this region, the GWL may exhibit a weak downward trend in the coming years. In most parts of the southern region, the GWL depth is expected to continue to decrease in the coming period. It should be noted that these trends in the groundwater change may persist in the future if the conditions for GWL formation do not change (or change only slightly).

## 4. Conclusions

This study reports a 50-year time series (1970–2020) of GWL fluctuations in the city of Odessa, Ukraine. Geostatistical and MF-DFA analyses were performed to study the spatial distribution and multifractal behaviour of the GWL fluctuations and predict trends for the coming period.

The geostatistical analysis provided the following results: Overall, the GWL varied more significantly during the last decade. During 1972–1983, the groundwater in different parts of the study area showed an overall change. The cone of the depression continued to expand to the eastern area of the central region, and the groundwater depths continued to increase. On the contrary, the groundwater recovered in the western area of the central region and in the central and southern areas of the southern region. The change in the GWL depths during this period was due to the weakening of the influence of human factors (continuous leaks from water-bearing utilities, various waste filtration reservoirs, activity by seaports, and large-scale mining) and the increasing influence of structural factors (the sea level, climate, topography, and geology). During 1984–1990, the GWL depths showed an overall change. The GWL depths in the eastern part of the central region increased, and they rebounded in the western part of this region and in the central parts of the northern and southern regions. During this period, the GWL depth variation may have been influenced mainly by structural factors. During 1991–2004, the cone of the depression continued to expand and the GWL depths continued to increase in all regions. The results show that, during 2005–2020, there was a slight decrease in the GWL depths in all regions of the study area. The GWL depth changes during 1991–2020 were caused by the weakening of the influence of structural factors and the increasing influence of human factors.

The MF-DFA results suggest that the dynamics of the GWL fluctuations have multifractal characteristics in the city area. The generalised Hurst exponent has a wide range of change of $0.20 < h(q) < 2.85$. From a hydrogeological perspective, this is expressed by the strong sensitivity of GWL fluctuations to changes in small-scale factors, including human factors (local leaks from water-bearing utilities, local heavy precipitation, etc.) and large-scale factors (the topography, geology, sea level, and climate, and large-scale human factors, such as activity by seaports and large-scale mining). The singularity spectrum

with the $\Delta\alpha$ was found in the limits of 0.23–2.60, except for three wells, for which the $\Delta\alpha$ was found to be 4.16, 4.26, and 4.47, which indicates a high degree of multifractality in the GWL time series and suggests that the groundwater multifractal behaviour is quite specific. For less than half of the wells, the time series demonstrate that both the long-range correlation and the fat-tail probability distribution contribute to the multifractality. We can conclude that the long-range correlations among the fluctuations are major contributors to the observed multifractality of the GWL fluctuation time series. Moreover, in most parts of the study area, the Hurst exponents ($q = 2$) are greater than 0.5, which indicates that the GWL fluctuations have a persistent behaviour and may show a stable trend. The Hurst exponent shows that the groundwater in most of northern region may show a slight upward trend in the near future. In the eastern part of the central region, the groundwater may show a downward trend in the near future. On the contrary, the groundwater in the remaining parts of the central region will probably maintain a continuous recovering trend. For most of the southern region, the groundwater is expected to continue a rebounding trend in the near future. It should be noted that these trends in the groundwater changes may persist in the future if the conditions of the GWL formation do not change (or change only slightly).

The GWL depth variation is affected by both natural factors (the climate, sea level, topography, aquifer lithology, and different hydrogeological units) and human factors (continuous leaks from water-bearing utilities, various waste filtration reservoirs, activity by seaports, and large-scale mining).

**Author Contributions:** All authors contributed to the conceptualisation and design of the study, and to the reading and revising of the manuscript. Methodology, D.M.; software, S.S and D.M.; investigation, D.M.; data curation, D.M.; writing—original draft preparation, D.M.; writing—review and editing, D.M. and S.S.; visualisation, D.M. and S.S. All authors have read and agreed to the published version of the manuscript.

**Funding:** This research received no external funding.

**Data Availability Statement:** The data used in this study are available from the Department of Engineering Geology and Hydrogeology (Mechnikov Odessa National University, Ukraine) upon reasonable request.

**Acknowledgments:** The authors acknowledge support from the dScience—Centre for Computational and Data Science—at the University of Oslo (UiO).

**Conflicts of Interest:** The authors declare no conflicts of interest.

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
