# Peer review of "Spatiotemporal Variation and Long-Range Correlation of Groundwater Levels in Odessa, Ukraine"

_water, doi:10.3390/w16010147_

Round 1

Reviewer 1 Report

Comments and Suggestions for Authors

The paper provides a comprehensive analysis of groundwater level (GWL) fluctuations in Odessa City, Ukraine, using geostatistical and multifractal detrended fluctuation analysis (MF-DFA) methods. The inclusion of both spatial and temporal analysis techniques adds depth to the study.

1. Methodology: The paper mentions the use of integrated time series and segmentation into non-overlapping windows. Can you provide more details on how the time series was integrated and the specific criteria used for window segmentation? 2. Trend Calculation: The paper mentions the use of a least square fit method for trend calculation. Can you explain the rationale behind choosing a first-order polynomial for local trend fitting? Were other polynomial orders considered? 3. Fluctuation Function: The paper describes the determination of the fluctuation function for different segments. Can you explain the significance of the parameter q and its relationship with the scaling behavior of the segments? 4. Scaling Behavior: The paper mentions the analysis of log-log plots to determine the scaling behavior of the fluctuation functions. Can you provide more details on how the generalized Hurst exponent h(q) is calculated and its interpretation in the context of stationary and non-stationary time series? 5. MF-DFA Method: The paper introduces the MF-DFA method for the multifractal characterization of nonstationary time series. Can you explain how this method differs from the traditional DFA method and its advantages in analyzing groundwater level fluctuations? 6. Data Analysis: The paper discusses the spatiotemporal variation of groundwater level depth in the Odessa City area. Can you provide more details on the data sources and the specific factors considered in the analysis, such as small-scale human factors and large-scale structural factors? 7. Results: The paper presents the spatiotemporal distribution of groundwater level depth and the semivariogram parameters. Can you explain the observed trends and patterns in the data, particularly the changes in spatial correlation and the influence of different factors? 8. Limitations: Are there any limitations or potential sources of error in the methodology or data analysis that should be considered? How do these limitations affect the interpretation of the results? 9. Future Research: Based on the findings of this study, are there any recommendations for future research or potential applications of the methodology in other areas or domains? 10. Conclusion: Can you summarize the main findings of the study and their implications for understanding groundwater level fluctuations in the Odessa City area?

Reviewer 2 Report

Comments and Suggestions for Authors

Dear Authors,

I received an article for review titled: "Spatiotemporal variation and long-range correlation of groundwater level in Odessa, Ukraine" regarding the assessment of changes in groundwater levels in Odessa, Ukraine. I read the article with great interest and commitment. The work refers to 46 items of literature, new and concerning issues related to the presented research. The structure of the article is correct and transparent. In my opinion, the study was designed and conducted correctly and described appropriately. My comments mostly concern the collected data and the presentation of figures in the paper:

1. I believe that the Introduction should delve deeper into the results achieved by other scientists - what exactly they studied, what they found, how similar it is to your research. In fact, the work simply mentioned that there were other studies but without their interpretation.

2. I think, introduction part should be ended with some research question, the thesis.

3. In the chapter 2 "Materials and methods"  - you should add some information about climate conditions and geology/hydrogeology of the researchred region.

4. In the chapter 2 "Materials and methods" - I did not find a description of wells you take into consideration in your research. I lack an explanation as to whether the wells were taken for testing were drilled or equipped with piezometer. I did not find an information about the depth of wells and reference used in the research. Did you used confined or unconfined water mirror? Was the groundwater level taken directly from measurements or was this level related to the ground surface?

5. Page 4, lines 135 and 142: why is the year 2000 referenced? After all, the data range was longer

6. Table 1 - Maybe instead of this table it would be more interesting to show the time series of all wells separately on one chart. On such a chart you can then see the amplitude of changes, seasonality, trends, and differences between individual time series from individual wells.

7. Figure 2 - the descriptions of the vertical and horizontal axes are much too small and therefore illegible. Does the chart show averages for all wells in each year?

8. Chapter 3 _results and discussion" - this chapter mostly gives raw data, without any possum or interpretation. I think it would be worth expanding the discussion

9. Figure 4 - the descriptions of the vertical  axes are much too small and therefore illegible.

Best regards:)

Round 2

Reviewer 1 Report

Comments and Suggestions for Authors

The revised manuscript has met the standards for publishing in Water.

Comments on the Quality of English Language

No  ang other suggestions.